# Thermodynamics-inspired high-entropy oxide synthesis

**Saeed S. I. Almishal** [1] ✉, **Matthew Furst** [1], **Yueze Tan** [1], **Jacob T. Sivak** [2], **Gerald Bejger** [3], **Joseph Petruska** [1], **Sai Venkata Gayathri Ayyagari** [1], **Dhiya Srikanth** [1], **Nasim Alem** [1], **Christina M. Rost** [3], **Susan B. Sinnott** [1,2,4], **Long-Qing Chen** [1] & **Jon-Paul Maria** [1]

High-entropy oxide (HEO) thermodynamics transcend temperature-centric approaches, spanning a multidimensional landscape where oxygen chemical potential plays a decisive role. Here, we experimentally demonstrate how controlling the oxygen chemical potential coerces multivalent cations into divalent states in rock salt HEOs. We construct a preferred valence phase diagram based on thermodynamic stability and equilibrium analysis, alongside a high throughput enthalpic stability map derived from atomistic calculations leveraging machine learning interatomic potentials. We identify and synthesize seven equimolar, single-phase rock salt compositions incorporating Mn, Fe, or both, as confirmed by X-ray diffraction and fluorescence. Energy-dispersive X-ray spectroscopy confirms homogeneous cation distribution, whereas X-ray absorption fine structure analysis reveals predominantly divalent Mn and Fe states, despite their inherent multivalent tendencies. Ultimately, we introduce oxygen chemical potential overlap as a key complementary descriptor for predicting HEO stability and synthesizability. Although we focus on rock salt HEOs, our methods are chemically and structurally agnostic, providing a broadly adaptable framework for navigating HEOs thermodynamics and enabling a broader compositional range with contemporary property interest.

High-entropy oxides (HEOs) redefine ceramics discovery by harnessing chemical disorder to unlock otherwise inaccessible chemistries through enthalpy-minimization approaches[1]. Despite HEOs' growing research body, much remains to be uncovered about the principles governing their stability and ability to form single-phase multi-component materials[1–5]. Addressing these knowledge gaps is essential to advancing HEO discovery and development. Undoubtedly, configurational entropy plays a critical role in stabilizing multicomponent solid solutions, especially at elevated temperatures where the thermal energy of mixing $(-T\Delta s_{mix}$; where $T$ is the temperature and $\Delta s_{mix}$ is the molar entropy of mixing which is dominated by configurational

entropy) rivals or exceeds the enthalpy of mixing per mol $(\Delta h_{mix})$ in minimizing the solid solution chemical potential $(\Delta\mu = \Delta h_{mix} - T\Delta s_{mix})$[2,3,6,7]. However, single-phase stability and synthesizability are not guaranteed by simply increasing configurational entropy; enthalpic contributions and thermodynamic processing conditions must also be carefully considered[1–9].

The Hume-Rothery rules, adapted and extended to ceramics, offer straightforward yet effective guidelines for predicting equilibrium solid solution formation based on enthalpic considerations[10]. The prototypical HEO, $Mg_{1/5}Co_{1/5}Ni_{1/5}Cu_{1/5}Zn_{1/5}O$ (MgCoNiCuZnO for brevity), does not comply with the Hume-Rothery crystal structure

[1]Department of Materials Science and Engineering, The Pennsylvania State University, University Park, PA, USA. [2]Department of Chemistry, The Pennsylvania State University, University Park, PA, USA. [3]Department of Materials Science and Engineering, Virginia Polytechnic Institute and State University, Blacksburg, VA, USA. [4]Institute for Computational and Data Sciences, The Pennsylvania State University, University Park, PA, USA. ✉e-mail: saeedsialmishal@gmail.com

compatibility criterion, as 2/5 cations prefer alternative crystal structures; ZnO favors the wurtzite structure, while CuO prefers the tenorite structure[1–3]. Therefore, both the high configurational entropy and the rock salt structure inherent stability, which possesses the broadest basin of attraction in binary oxides[11], drive the single-phase rock salt solid solution formation. MgCoNiCuZnO adheres closely, however, to every other Hume-Rothery criterion, including the cation radii, electronegativity, and valence compatibility. The largest size disparity occurs between $Ni^{2+}$ and $Co^{2+}$, with $Co^{2+}$ being 8% larger than $Ni^{2+}$ – still within the 15% Hume-Rothery limit. This criterion explains why Ca, Sr, or Ba cannot be incorporated in equimolar amounts into MgCoNiCuZnO under equilibrium synthesis conditions[8,12]. Finally, MgCoNiCuZnO stability is also intrinsically tied to its cations favoring and maintaining a 2+ oxidation state in their binary oxides over the critical 875–950 °C temperature range for phase stabilization under ambient oxygen partial pressure (ambient $pO_2$), along with minimal electronegativity variation among the 2+ cations[13]. This valence compatibility criterion explains why incorporating persistent $Sc^{3+}$ into equimolar rock salt HEOs is challenging, if not impossible, under equilibrium synthesis conditions, despite its ionic radius being comparable to other cations in MgCoNiCuZnO[14].

To expand the rock salt high entropy oxide library via near equilibrium routes, we therefore need to identify cations whose ionic radii closely match MgCoNiCuZnO average radius and that can be coerced to take a 2+ oxidation state. Unlike Sc, all other 3 d transition metals, including and beyond those in MgCoNiCuZnO, can adopt a 2+ oxidation state. However, their stability in the 2+ state depends on their electronic configuration as well as their thermodynamic and kinetic processing factors. Among those, Mn is positioned at the 3d-period center with five unpaired electrons, leading to the largest and highest possible oxidation states in the entire period. Therefore, Mn exhibits remarkable versatility, forming diverse oxide structures and phases. Under ambient atmospheric pressure, Mn commonly exists as tetragonal pyrolusite ($MnO_2$)[15,16]. However, at elevated temperatures, such as those used in our HEO synthesis (~460 °C and above), Mn transitions

to $Mn_2O_3$ typically adopting the bixbyite or corundum structures[15,16] (Supplementary Information Note 1). Following Mn in the periodic table is Fe, which is stable as $Fe_2O_3$ in the hematite phase (corundum structure) under ambient conditions – unlike Mn, Fe is largely restricted to the 2+ and 3+ oxidation states. Importantly, both Mn and Fe can be readily coerced into the 2+ oxidation state under laboratory-accessible reducing conditions. In contrast, while Ti, V, and Cr can also adopt 2+ oxidation states, they require extreme reducing conditions compared to all other cations in the 3 d period (Supplementary Information Note 6).

Therefore, Mn and Fe are compelling candidates for incorporation into rock salt HEOs, as they can adopt a 2+ oxidation state and maintain size compatibility in both their 2+ and 3+ states, with ionic radii deviating within 15% of the cation radii in MgCoNiCuZnO. While entropy stabilization is typically explored through cation selection, widely accepted ambient $pO_2$, and high temperature, we establish the oxygen chemical potential as a powerful yet underutilized thermodynamic axis for controlling phase stability. By precisely tuning $pO_2$ during synthesis, we suppress higher oxidation states and promote $Mn^{2+}$ and $Fe^{2+}$ incorporation. To harness this control in the simplest manner, we propose starting with AO oxide mixtures and employing high-temperature synthesis under a controlled, continuous Argon (Ar) flow to maintain low $pO_2$, effectively steering different compositions toward a stable, single-phase rock salt structure.

## Results and discussion
### Rock salt high-entropy oxide thermodynamics
We begin by examining fundamental thermodynamic variables to assess rock salt HEOs stability and develop intuition into their synthesizability. In Fig. 1a, we present an enthalpic stability map (adapted from our study in ref. 8); with mixing enthalpy ($\Delta H_{mix}$) in meV/atom and bond length distribution ($\sigma_{bonds}$) in Å as its axes[8]. $\Delta H_{mix}$ represents the enthalpic barrier to single-phase formation, while $\sigma_{bonds}$ quantifies lattice distortion through the relaxed first-neighbor cation-anion bond lengths standard deviation (see Methods). A lower $\sigma_{bonds}$ suggests

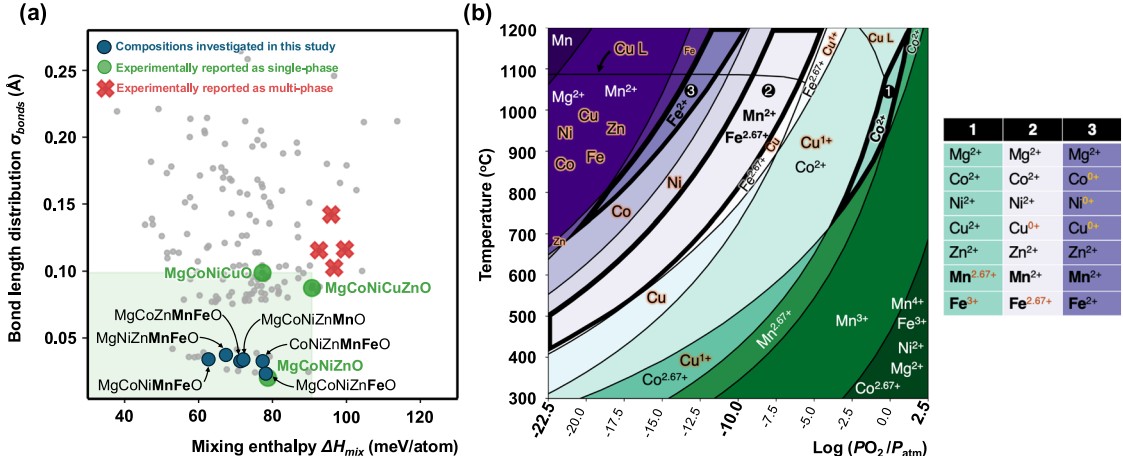

**Fig. 1 | Composition and thermodynamic landscape governing stability in rock salt high-entropy oxides. a** Rock salt HEO composition map with $\Delta H_{mix}$ and $\sigma_{bonds}$. The map includes all equimolar four-, five- and six-component systems drawn from the cation cohort: Mg, Ca, Mn, Fe, Co, Ni, Cu and Zn as gray points. Experimental synthesis results for single and multi-phase are indicated with green circles and red crosses, respectively. Region predicted as single-phase are shown in a light green shade. All 5-component compositions in this study are labeled and indicated with blue circles. Gray points with descriptor values lower than those of the five-component systems in blue circles correspond to four-component compositions (majority without Zn), as reported in Table S3. Figure 1a and its datasets are adapted with permission from J.T., Sivak, S.S.I. Almishal et al., Phys. Rev. Lett. 134, 216101 (2025). Copyright (2025) by the American Physical Society. **b** Temperature and

oxygen partial pressure phase diagram illustrating each cation preferred oxidation state in its stable binary oxide phase. In region (1) all cations in $Mg_{1/5}Co_{1/5}Ni_{1/5}Cu_{1/5}Zn_{1/5}O$ are stable in 2+ oxidation state in their $A^{2+}O^{2-}$ binary oxide form; in region (2) all cations in $Mg_{1/5}Mn_{1/5}Co_{1/5}Ni_{1/5}Zn_{1/5}O$ are stable in 2+ oxidation state in their $A^{2+}O^{2-}$ binary oxide form; and in region (3) Fe is stable in the 2+ oxidation state in its $A^{2+}O^{2-}$ binary oxide form, while Ni and Co reduce to their metallic states. Orange shading indicates cations reduced to oxidation states lower than 2 + . Because Ti, V, and Cr require substantially lower oxygen partial pressures to maintain their 2+ oxidation states, we excluded them from this analysis (see Supplementary Information Note 6). The uncolored version of (**b**), with legends, is included in the Source Data file.

minimal lattice distortion, promoting single-phase stability analogous to the Hume-Rothery ionic size rule. To construct and populate this stability map, we leverage the Crystal Hamiltonian Graph Neural Network (CHGNet) machine learning interatomic potential, which achieves near-density functional theory accuracy with considerably reduced computational cost[17]. The map includes all equimolar four-, five- and six-component compositions drawn from the cation cohort: Mg, Ca, Mn, Fe, Co, Ni, Cu and Zn (Supplementary Information Note 7 Table S3). It stands out that all five-component compositions containing Mn and Fe, but lacking Ca and Cu (denoted as blue circles in the map) exhibit the lowest $\Delta H_{mix}$ and $\sigma_{bonds}$ values among all the other five-component compositions (all numerical values can be found in Supplementary Information Note 7 Table S3), with values even lower than the prototypical MgCoNiCuZnO[8,18,19]. Despite these favorable characteristics, Mn and Fe-based compositions have eluded conventional synthesis routes for the past decade[2,19,20]. MgCoNiMnFeO is the only composition among those highlighted in the map that has been experimentally reported[19]. Pu et al. synthesized it via an elaborate method starting from oxalate precursors followed by annealing under a controlled atmosphere that involved both reducing and oxidizing agents[19]. Notably, this same composition, MgCoNiMnFeO, has the lowest $\Delta H_{mix}$ and $\sigma_{bonds}$ among the six five-component compositions that exclude Ca and Cu. This arises because it lacks Zn, which preferentially stabilizes in a wurtzite – rather than a rock salt – structure in its oxide form. Consequently, to our best knowledge, no equilibrium-synthesized rock salt HEOs containing Mn, Fe, and Zn have been reported.

To reconcile the low $\Delta H_{mix}$ and $\sigma_{bonds}$ values for the Mn and Fe-containing compositions with their synthesis challenges, we use CALPHAD to construct a temperature–oxygen partial pressure phase diagram (Fig. 1b). This diagram maps the stable Mg and 3 d transition metal (Mn through Zn) oxidation states in their binary oxide phases and delineates temperature-pressure zones where their valence stability windows partially or fully overlap. We focus on three distinct regions, each designated by a number and outlined in bold, with the stable valence of each cation in these regions listed in the accompanying side table. The phase diagram reveals that under ambient conditions (the far bottom-right region), Mn predominantly adopts a 4+ oxidation state, Fe adopts a 3+ state, Co averages 2.67+ (due to mixed valencies), and the remaining cations persist in the 2+ state. In contrast, at extremely low oxygen partial pressures (~$10^{-15}$–$10^{-22.5}$ bar) and temperatures above ~800 °C, all cations except Mg eventually reduce to their metallic forms. In Region 1 (ambient pressure, T > ~875 °C), only the cations in prototypical MgCoNiCuZnO are stable in their $A^{2+}O^{2-}$ binary oxide phases (see the side table), explaining this composition's unique stability under ambient conditions. Deviations from Region 1, either toward higher temperatures or lower $p$O₂, inevitably drive CuO reduction and Cu melting under equilibrium synthesis methods, with the Cu liquidus line being the only liquidus boundary in Fig. 1b. Absent Cu, as $p$O₂ decreases from Region 1, Mn reduces to 2 +, marking the transition into Region 2, whereas further reductions stabilize $Fe^{2+}$, defining Region 3, where importantly, Mn remains 2+ stable. Therefore, Regions 2 to 3 outline the synthesis conditions under which rock salt high-entropy oxides containing Mn and Fe, but not Cu, can be stabilized based solely on oxidation-state compatibility criteria. We therefore experimentally explore incorporating Mn and Fe within the rock salt high-entropy oxide family by first synthesizing the six five-component Mn- and Fe-containing compositions identified in the stability map in Fig. 1a and maintaining low $p$O₂ values to access regions 2 to 3 in Fig. 1b. The six compositions are as follows: first, Mn is introduced into MgCoNiZnO (a four-component derivative of the prototypical MgCoNiCuZnO obtained by removing CuO) forming MgCoNiZnMnO. Next, Mn is replaced with Fe, yielding MgCoNiZnFeO. Finally, both Mn and Fe are incorporated simultaneously, each time substituting one of the four cations in MgCoNiZnO,

resulting in four additional compositions: MgCoNiMnFeO, MgNiZnMnFeO, CoNiZnMnFeO, and MgCoZnMnFeO.

## Synthesizing Mn- and Fe-containing rock salt high-entropy oxides

We hypothesize all six Mn- and Fe-containing five-component rock salt phases shown in blue in Fig. 1a can be stabilized by combining $A^{2+}O^{2-}$ binary oxides and processing them at high temperatures under a low $p$O₂ environment. This approach suppresses higher oxidation states, ensuring that the cation valence states remain consistent with Regions 2 to 3 in Fig. 1b. To achieve this low $p$O₂ environment in practice, we react and sinter these oxides in a tube furnace under Ar flow, as illustrated in the experimental setup in Fig. 2. Figure 3 presents the X-ray diffraction (XRD) patterns for these Mn- and Fe-containing compositions, compared to the prototypical MgCoNiCuZnO. All the samples are sintered at 1100 °C for 5 h: one set in air for comparison (Fig. 3a) and the other in Ar, as illustrated by the schematic in Fig. 2, to validate our hypothesis (Fig. 3b). Figure 3a confirms that all the Mn and Fe-containing compositions processed in air form a spinel phase in addition to the rock salt phase – despite all the starting precursors being in their $A^{2+}O^{2-}$ binary oxide form. This underscores Mn and Fe tendencies for higher oxidation states at ambient $p$O₂ in agreement with the phase diagram in Fig. 1b (see also Supplementary Information Note 1). In comparison, when we restrict the oxygen availability to that provided by the starting oxide precursors by processing our compositions in Ar, the resulting XRD patterns in Fig. 3b confirm that all the Mn- and Fe-containing compositions predominantly form a single-phase rock salt structure, supporting our arguments and rationale. In other words, flowing only Ar produces $p$O₂ akin to Regions 2 to 3 in Fig. 1b (see Methods). Combined with the AO starting powders and configurational entropy contributions, this leads to forming single-phase high-entropy rock salts. Notably, the prototypical MgCoNiCuZnO develops a reduced metallic phase after high-temperature processing in Ar, which we attribute to Cu reduction and melting. Figure 3c presents the X-ray fluorescence (XRF) spectra for all the compositions corresponding to those in Fig. 3b, confirming both their composition and closely equimolar stoichiometry (see Table S1, Supplementary Information Note 2 for the details).

To confirm that Mn and Fe each maintain a 2+ valence within the HEO matrix, we measure the X-ray absorption fine structure (XAFS) spectra with only the X-ray absorption near edge (XANES) region of CoNiZnMnFeO and MgNiZnMnFeO; both compositions contain Mn, Fe, Ni, Zn, and either Co or Mg. We choose these compositions as they are among the most challenging to stabilize (Fig. 1a), as one lacks Co or Mg: key rock salt stabilizers under our synthesis conditions, and both include Zn, whose wurtzite preference as an oxide adds structural and enthalpic complexity[3]. The rising absorption edge in Fig. 4a, c, called the white line, represents electronic transitions that appear as inflection points, which the spectrum's derivative precisely locates. Here, E₀, the edge photon energy, is defined as the first main peak in the first derivative (Supplementary Information Note 3), excluding contributions from pre-edge features. The E₀ values for 3d-transition metal K-edges increase linearly with the valence states[21], enabling a best-fit line to estimate unknown valence states, as we indicate in Figs. 4b, d for Mn and Fe, respectively (see Supplementary Information Note 3, Table S2 for numerical values). Using this approach, all the measured cations in MgNiZnMnFeO and CoNiZnMnFeO, including Mn and Fe, predominantly exhibit a 2+ valence state. This is evident from the proximity of their measured E₀ edge energies to those corresponding to 2+ from reference samples (indicated by the orange stars in Fig. 4b, d).

## The parent six-component composition: MgCoNiZnMnFeO

To further solidify our findings, we strategically extend our established thermodynamic analysis and synthesis to a seventh composition which

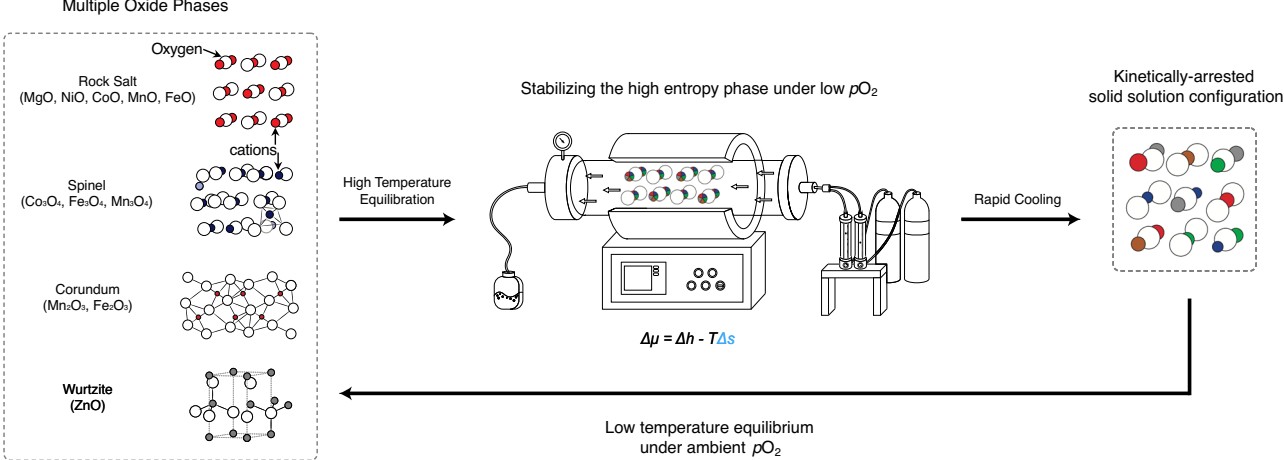

**Fig. 2 | Mn- and Fe-containing rock salt compositions evolution with temperature.** The flowchart illustrates the phase progression in the Mn- and Fe-containing compositions with temperature, emphasizing that stabilizing these compositions as single-phase rock salt requires synthesis under a controlled atmosphere. White circles represent oxygen, while colored circles indicate cations (shown with reduced connectivity in projection for simplicity; e.g., corundum cations are octahedrally coordinated though only four bonds are immediately visible here). We highlight that in addition to wurtzite ZnO, spinel phases ($Co_3O_4$, $Mn_3O_4$ and $Fe_3O_4$) and corundum phases ($Mn_2O_3$ and $Fe_2O_3$) should be explicitly considered in low temperature processes as competing phases. Note $Mn_2O_3$ can also form bixbyite $Ia\bar{3}$ phase and $Mn_xFe_yO_\delta$ sintered in air forms a spinel and bixbyite mixture, as detailed in Supplementary Information Note 1. $\Delta\mu$ is the change in chemical potential, $\Delta h$ is the change in molar enthalpy, $T$ is temperature, and $\Delta s$ is the change in molar entropy. This figure is expanded from our original figure in G.N.K Kotsonis, S.S.I. Almishal et al., Journal of the American Ceramic Society, 106(10), 5587−5611 [1].

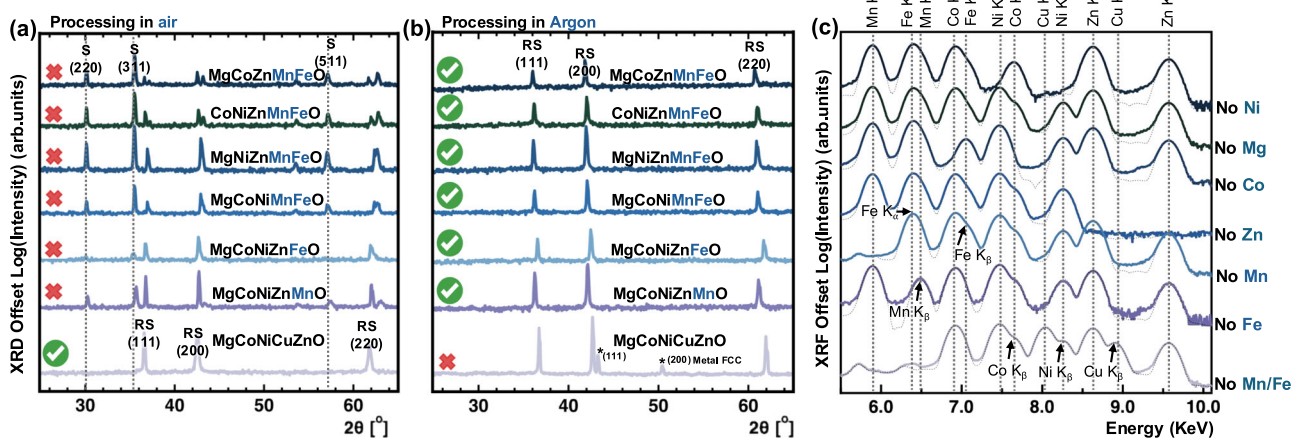

**Fig. 3 | Structural and chemical characterization of Mn- and Fe-containing rock salt high-entropy oxides. a** X-ray diffraction scans of prototypical single-phase MgCoNiCuZnO and the six five-component systems containing Mn, Fe or both after firing in ambient $pO_2$. All systems containing Mn and Fe form a spinel and rock salt phase mixture. 'RS' denotes the peaks corresponding to the rock salt structure, while 'S' denotes those corresponding to the cubic spinel structure. Red crosses indicate compositions that did not form the desired rock salt phase, while green check marks denote successful stabilization of the rock salt structure. **b** X-ray diffraction scans of the same systems under processing in a reducing environment with Ar gas flow. All systems containing Mn and Fe form a single-phase rock salt, while MgCoNiCuZnO forms secondary phases as a result of excess reduction. **c** X-ray fluorescence spectra of each composition highlighting all detected cations except Mg. Thick colored spectra represent the measured data, while thin dotted lines indicate the corresponding fits generated by the calibrated quantification software. Fitting details and elemental concentrations are provided in Supplementary Information Table S1. Source data are provided as a Source Data file.

is the parent six-component composition $Mg_{1/6}Co_{1/6}Ni_{1/6}Zn_{1/6}Mn_{1/6}Fe_{1/6}O$ (MgCoNiZnMnFeO), which unifies all our targeted cations and serves as an ideal platform for rigorous characterization. It is reassuring that MgCoNiZnMnFeO exhibits a 71 meV/atom $\Delta H_{mix}$ and a 0.036 Å $\sigma_{bonds}$ in Fig. 1a, which are comparable to its five-component derivatives. We start by synthesizing it under the same conditions that successfully stabilize the six five-component rock salt HEO phases in Fig. 3b. We then examine its synthesizability under more stringent reducing conditions to further examine our thermodynamic framework predictive capability. Finally, we perform detailed structural and chemical characterization on the sample synthesized under our ideal

conditions, probing the structure, elemental homogeneity, and Mn and Fe oxidation states using transmission electron microscopy (TEM) selected area electron diffraction (SAED), energy dispersive spectroscopy (EDS), and XANES, respectively.

Figure 5a presents the XRD patterns of MgCoNiZnMnFeO for the same pellet sintered consecutively at different temperatures, each for 5 h under a 100 SCCM Ar flow. Up to 850 °C, a rock salt phase predominates, with wurtzite being the only secondary phase which likely corresponds to partially-unreacted ZnO phase. Beyond that temperature, the XRD pattern indicates a single-phase rock-salt structure with notably narrower peaks at 1100 °C, further supporting our reasoning

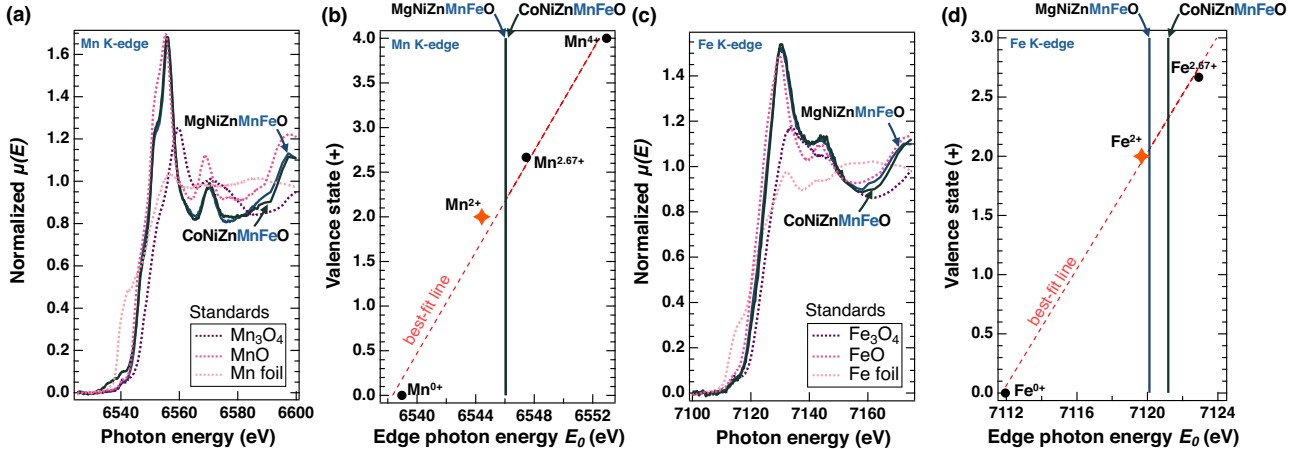

**Fig. 4 | Local electronic structure analysis confirms 2+ oxidation state in Mn- and Fe-containing compositions. a** X-ray absorption near edge structure (XANES) measurements of Mn K-edge spectra in MgNiZnMnFeO and CoNiZnMnFeO in comparison to Mn$^{x+}$ reference spectra from standards. **b** Mn K-edge photon energy versus valence state with best-fit line confirming a predominance of Mn$^{2+}$ within both high entropy compositions, the 2+ reference value is indicated by the orange

star. **c** XANES measurements of Fe K-edge spectra in MgNiZnMnFeO and CoNiZnMnFeO in comparison to Fe$^{x+}$ reference spectra from standards. **d** Fe K-edge photon energy vs valence state with best-fit line confirming a predominance of Fe$^{2+}$ within both high entropy compositions, the 2+ reference value is indicated by the orange star. Source data are provided as a Source Data file.

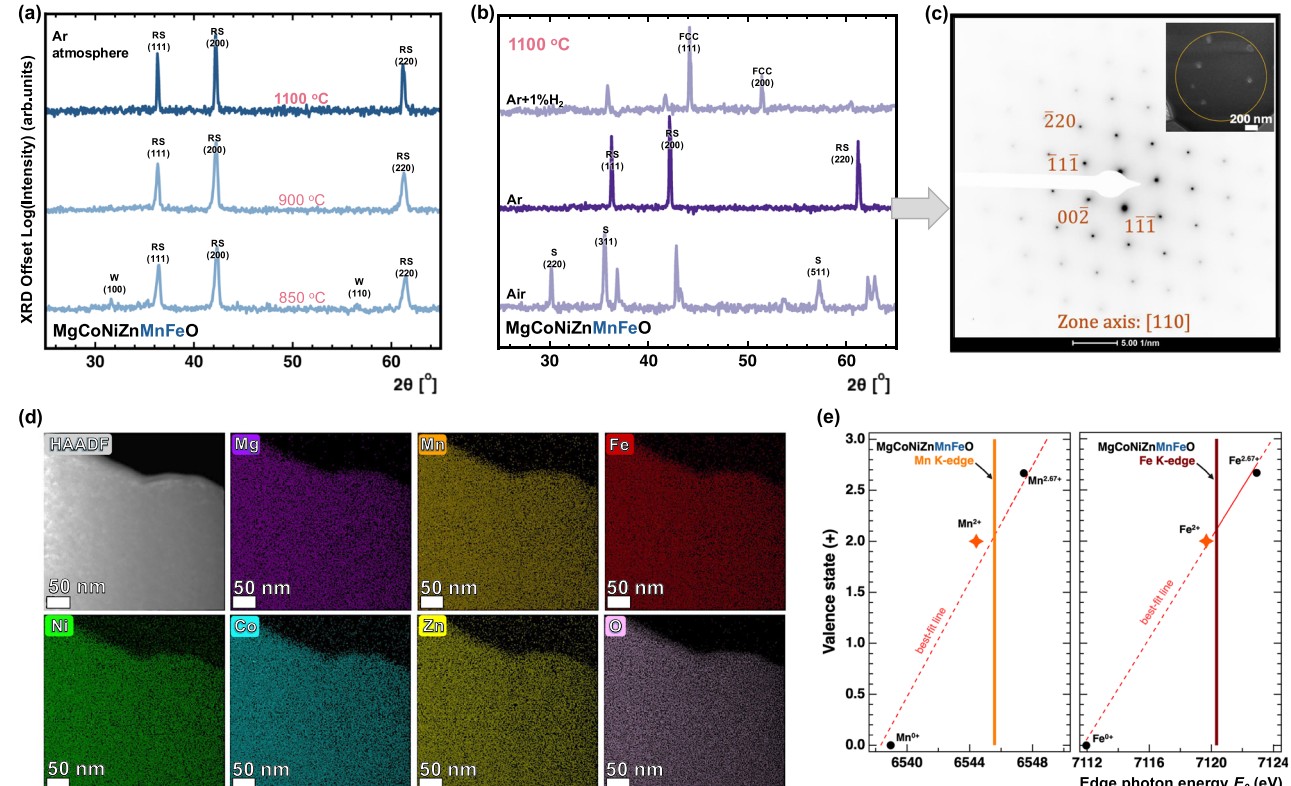

**Fig. 5 | Phase evolution, structural and chemical characterization, and elemental homogeneity in the six-cation MgCoNiZnMnFeO parent high-entropy oxide. a** X-ray diffraction patterns of the 6-component high-entropy composition Mg$_{1/6}$Co$_{1/6}$Ni$_{1/6}$Zn$_{1/6}$Mn$_{1/6}$Fe$_{1/6}$O (MgCoNiZnMnFeO) sintered for 5 h under 100SCCM of Ar at different temperatures, suggesting that the transition to single phase occurs between 850–900 °C with the disappearance of the wurtzite peaks. 'RS' denotes peaks from the rock salt structure, and 'W' indicates the wurtzite structure. **b** X-ray diffraction patterns of MgCoNiZnMnFeO sintered at 1100 °C under varying oxygen partial pressures. A single-phase rock salt structure forms after 5 h under a 100 SCCM Ar flow. In contrast, sintering in air results in the emergence of a spinel phase, while introducing a small percentage of H$_2$ leads to the formation of a reduced metallic phase. 'RS' denotes peaks from the rock salt

structure, 'S' denotes those from the cubic spinel structure, and 'FCC' indicates the face-centered cubic metal structure. **c**–**e** are obtained by characterizing the parent composition MgCoNiZnMnFeO sintered for 5 h under 100SCCM Ar. **c** Selected area electron diffraction (SAED) along the [110] zone axis of MgCoNiZnMnFeO sintered for 5 h under 100SCCM Ar, with the inset showing the selected area (yellow circle ~700 nm in radius) corresponding to the diffraction pattern. The electron diffraction pattern is consistent with the rock salt crystal structure. **d** Energy-dispersive spectroscopy (EDS) maps showing a homogeneous distribution of elements at the 50 nm scale. **e** Mn K-edge and Fe K-edge photon energy vs valence state with best-fit line confirming a predominance of Mn$^{2+}$ and Fe$^{2+}$ within MgCoNiZnMnFeO composition, with the 2+ reference values are indicated by orange stars. Source data for (**a**), (**b**) and (**e**) are provided as a Source Data file.

and methods (see Supplementary Information Note 2 for all compositions' wide $2\theta-\theta$ scans that we stabilize in the single-phase rock salt structure). In Fig. 5b, we present three distinct MgCoNiZnMnFeO ceramic pellets that are prepared and sintered at 1100 °C for 5 h, each in a different atmosphere: air (ambient $pO_2$), 100 SCCM Ar, and 100 SCCM forming gas (99% Ar + 1% $H_2$) (details in Methods). The Ar-sintered sample forms a stable single-phase rock-salt structure, consistent with our previous conclusions. The air-sintered sample is predominantly spinel, and the forming gas sample is partially reduced to metallic phase(s). Notably, an additional 1% $H_2$ lowers $pO_2$ to that in Region 3 in Fig. 1b or beyond. Therefore, observing those metallic phases follows exactly the valence phase diagram in Fig. 1b, which shows that FeO formation conditions at 1100 °C also reduce Co and Ni to their metallic states (see Supplementary Information Note 4 for another example demonstrating how metallic phases evolve under forming gas in the five-component MgCoNiMnFeO).

We next examine the sample synthesized under our ideal conditions (middle pellet, Fig. 5b) through detailed structural and chemical characterization, as shown in Fig. 5c–e. The TEM SAED pattern (Fig. 5c) confirms rock salt structure formation, with no additional diffraction peaks that could indicate secondary phases. The EDS maps (Fig. 5d) reveal a homogeneous, random cation distribution at the 50 nm scale, with similar observations at the 100 nm and 500 nm scales (Supplementary Information Note 5). XANES spectral fitting (consistent with the analysis approach in Fig. 4, with the spectra provided in Supplementary Information Note 3, Fig. S5) confirms that Mn and Fe are predominantly in the 2+ oxidation state, as shown in Fig. 5e. Together, the results in Fig. 5 confirm that a homogeneous, chemically disordered, positionally ordered single-phase rock salt that is decorated primarily by divalent cations. Importantly, without controlled near-equilibrium synthesis or rapid quenching, Mn, Co, and Fe are prone to multivalency, producing complex valence states and micro- to nanoscale structural diversity, as seen in related systems[5,9,22] (more discussion in Supplementary Information Note 9).

## Chemical potential overlap as a synthesis descriptor

Finally, we develop a simple synthesizability descriptor that both captures the key trends observed in the valence phase diagram (Fig. 1b) and can be readily extended to other systems at minimal cost. Although our experimental results thus far have focused on relatively high-temperature processing, valuable insights can still be gained from

ground-state (0 K) density functional theory (DFT) calculations, provided that the cation valence states are well represented[3,18,19]. The open-source Materials Project database offers powerful tools for this purpose, including the Phase Diagram and Chemical Potential Diagram modules[23,24]. Developed based on the pioneering work by Yokokawa et al.[25] and detailed by Todd et al.[26], these diagrams map each component's chemical potential onto an axis, delineating different stoichiometries stability regions. When choosing oxygen as a chemical potential axis, the diagram maps different oxidation states on the convex hull. For an arbitrary oxide $A_xO_y$, the y-axis can therefore represent the oxygen chemical potential, while the x-axis corresponds to cation A chemical potential, providing oxygen chemical potential regions where specific cation oxidation states are enthalpically stable. The Mg-O, Cu-O and Mn-O chemical potential diagrams are illustrated in Fig. 6a, and the diagrams for all 3 d transition metals can be found in Supplementary Information Note 6. Figure 6a shows an overlap in oxygen chemical potential for $Mg^{2+}$ and $Cu^{2+}$ as well as for $Mg^{2+}$ and $Mn^{2+}$; however, a separation exists between $Cu^{2+}$ and $Mn^{2+}$ along the oxygen chemical potential axis, in agreement with the valence phase diagram in Fig. 1b which shows that $Cu^{2+}$ and $Mn^{2+}$ do not coexist. We quantify this oxygen chemical potential overlap ($\mu_{overlap}$) for all 21 possible $A_{1/2}A'_{1/2}O$ cation combinations in Fig. 6b, rapidly evaluating common valence stability windows with minimal computational cost. A larger overlap (indicated in blue shade) suggests a wide stability window where cations coexist in their $A^{2+}O^{2-}$ stoichiometry, making single-phase synthesis feasible provided that this window can be accessed experimentally. In contrast, a greater separation (indicated in red shade) suggests a narrower stability window, making synthesis increasingly difficult or even impossible. Notably, this descriptor is impartial to the equilibrium crystal structure as long as it hosts the AO stoichiometry. The values for $\mu_{overlap}$ in Fig. 6b largely align with the understanding gained from Fig. 1b, such as $Mg^{2+}$ stability across a wide oxygen chemical potential region, whereas $Cu^{2+}$ with $Mn^{2+}/Fe^{2+}$ mixtures are likely not stable.

To provide a more comprehensive view for HEO synthesis, we extend the $\mu_{overlap}$ descriptor in Fig. 6c to all five- and six-cation combinations as a color overlay (Fig. 6c) in addition to our original descriptors $\Delta H_{mix}$ and $\sigma_{bonds}$ (Fig. 1a) (details in Methods and Supplementary Information Note 7). We use the same color scheme as in Fig. 6b to maintain visual consistency. Figure 6c suggests that despite MgCoNiCuZnO's relatively high $\Delta H_{mix}$ and $\sigma_{bonds}$ compared to other

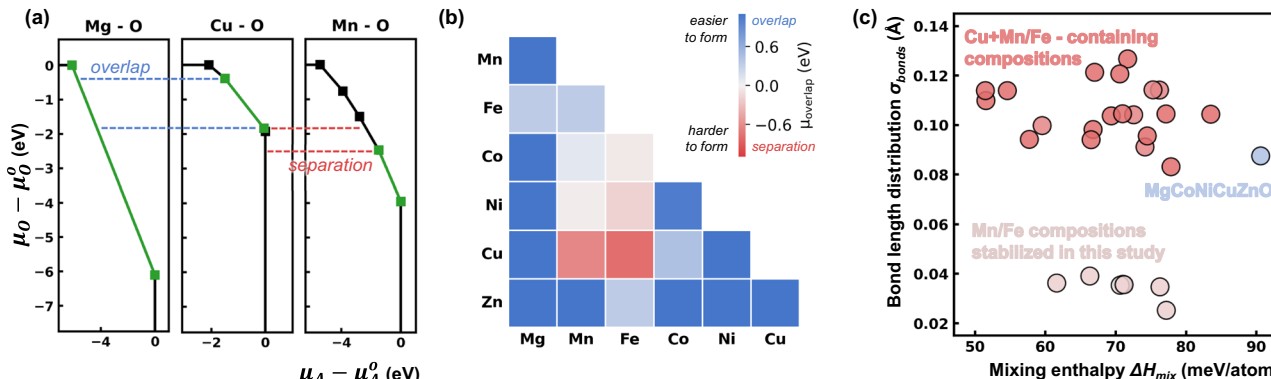

**Fig. 6 | Chemical potential overlap as a descriptor for phase stability and synthesizability in rock salt high-entropy oxides. a** Chemical potential diagrams for Mg-O, Cu-O and Mn-O extracted from the Materials Project database, demonstrating stability windows of each cation in their $A_xO_y$ binary oxides. A green color denotes oxygen chemical potential regions in which $A^{2+}O^{2-}$ compositions are stable. Note the significant overlap MgO has with CuO and MnO, but the separation between CuO and MnO. **b** Chemical potential overlap descriptor ($\mu_{overlap}$) for all two-cation AO combinations. Cu, Mn, and Fe consistently have the smallest $\mu_{overlap}$

with negative values corresponding to large separation, while Mg and Zn have the largest positive overlap. **c** Introducing chemical potential overlap as a third descriptor (color overlay) along with $\Delta H_{mix}$ and $\sigma_{bonds}$ reveals the uniqueness of prototypical MgCoNiCuZnO as the only 5- or 6-cation combination with significant overlap in $A^{2+}O^{2-}$ binary oxide stability windows. This diagram also demonstrates the difficulty of stabilizing compositions with Cu, Mn and Fe. The combinations are colored with the same scale in part (b). Source data for (**b**) and (**c**) are provided as a Source Data file.

compositions, it readily forms a single-phase rock salt AO structure as all its cations share an overlapping oxygen chemical potential window. We note that this overlap must coincide with ambient experimental conditions as MgCoNiCuZnO is the only rock salt composition explored that forms a single-phase in ambient $pO_2$ (Fig. 3a). Conversely, while compositions with Mn and Fe appear easiest to stabilize based on their low $\Delta H_{mix}$ and $\sigma_{bonds}$, the $\mu_{overlap}$ descriptor resolves the greater difficulty in stabilizing these compositions compared to MgCoNiCuZnO. Their narrow $\mu_{overlap}$ windows also lie lower on the oxygen chemical potential y-axis, indicating that more reducing conditions are required for synthesis (see Supplementary Information Note 7 for details on how the $\mu_{overlap}$ window position along the y-axis informs synthesis design). Finally, compositions containing Cu and Mn/Fe, despite their low $\Delta H_{mix}$, exhibit increased $\sigma_{bonds}$ and a large separation in chemical potential mainly due to Cu and Mn/Fe, making their stabilization more challenging, if not impossible.

While we focus here on the $A^{2+}O^{2-}$ stoichiometry for rock salt HEOs, this concept is similarly applicable to other chemistries and crystal systems. For example, in Supplementary Information Note 8, we extend this descriptor to rare-earth sesquioxides and defective fluorites, examining $(Ce,La,Pr,Sm,Y)_2O_{3+\delta}$ and $(Zr,Ce,La,Pr,Sm,Y)O_{2-\delta}$ as case studies. We can also apply the same framework to structures with multiple cation sublattices, such as ternary perovskites[9], by independently treating each site according to its preferred valence. Therefore, for accurate HEO stability predictions, it is critical to consider $\mu_{overlap}$ alongside conventional descriptors[8,18]. Specifically, a comprehensive predictive framework for realizing equilibrium HEO phases should, at a minimum, incorporate $\Delta H_{mix}$, $\sigma_{bonds}$ and $\mu_{overlap}$, together with insights from binary and higher-order phase diagrams and decomposition enthalpy[3,8]. Future studies should incorporate finite-temperature and configurational entropy effects, moving beyond the ideal mixing assumption, further enhancing stability and synthesizability predictions.

Overall, the enthalpic parameters $\Delta H_{mix}$ and $\sigma_{bonds}$ significantly advance predicting stable HEOs by capturing enthalpic barriers and lattice distortion. Alone, however, they suggest that Mn- and Fe-containing systems should stabilize more easily than the prototypical MgCoNiCuZnO. Yet Mn and Fe favor the 3+ oxidation state, at typical firing temperatures and ambient $pO_2$, often forming higher-valence phases like spinels. Positioning $pO_2$ as an indispensable axis in phase diagrams, and introducing $\mu_{overlap}$ as a third HEO screening descriptor, illuminates MgCoNiCuZnO's uniqueness; despite its higher $\Delta H_{mix}$ and $\sigma_{bonds}$ values compared to other five-component compositions, each cation remains in the 2+ state under our synthesis temperature and earth's ambient $pO_2$. In contrast, Mn- and Fe-containing compositions require reduced $pO_2$ to stabilize them in the rock salt phase. By optimizing the synthesis conditions to an overall lower $pO_2$, we successfully synthesize seven single-phase rock salt HEOs, bypassing the need for complex precursor chemistry. Explicitly introducing oxygen chemical potential into equilibrium descriptors recasts HEO discovery from empirical trial-and-error to valence-conscious predictive synthesis, establishing a broadly adaptable framework for realizing uncharted, functionally promising chemically disordered complex oxides.

## Methods

### Bulk synthesis
MgO (Sigma-Aldrich, 342793), CoO (Sigma-Aldrich, 343153), NiO (Sigma-Aldrich, 203882), CuO (Alfa Aesar, 44663), ZnO (Sigma-Aldrich, 96479), MnO (Sigma-Aldrich, 377201), and FeO (Sigma-Aldrich, 400866) starting precursors adopt the rock salt crystal structure, as confirmed by XRD. Figure S1a shows representative XRD patterns for the as-received MnO and FeO powders, both exhibiting the expected rock salt structure with no detectable secondary phases. To avoid potential oxidation or moisture uptake, we vacuum-seal the precursors

in inert packaging (VacMaster PRO350) and store them under ambient conditions. Within the resolution limits of our XRD measurements, we did not observe any structural degradation or secondary phase formation during storage.

Depending on the composition, we combine equimolar amounts of these precursors and homogenize the powders by speed mixing them in a FlackTek PP container using a FlackTek SpeedMixer at 2000 rpm for 1 min. After mixing, we transfer the powders to a Thermo Scientific amber wide-mouth HDPE bottle containing six 5-mm yttria-stabilized zirconia milling media. We then shake mill the mixture using a SPEX 8000 Mixer/Mill for 2.5 h. Finally, we press the milled powders into pellets 1.27 cm or 2.5 cm in diameter at 60–100 MPa using a Carver Laboratory Press, holding the pressure for 30 s. For the ideal conditions that produce high-quality single-phase rock salt structure in Mn and Fe containing compositions, we load the pressed pellets onto custom-made alumina boats (AdValue Technology) and place them in a 5 cm diameter quartz tube furnace (Across International NC2156188). We seal the furnace and flow 100 SCCM of Linde UHP Ar, controlled by a Brooks mass flow controller. Flowing 100 SCCM Ar at 1100 °C in our setup yields $pO_2$ in the range of $10^{-6}$–$10^{-8}$ atm. After stabilizing the gas flow, we ramp the temperature from room temperature to 1100 °C over 35 min and hold it at 1100 °C for 5 h. After the hold, we begin cooling still under continuous Ar flow, decreasing the temperature at 40 °C/min down to 700 °C. We then open the tube furnace insulation while maintaining the 100 SCCM Ar flow and place two 15 W NMB-MAT cooling fans at both ends of the tube. This setup rapidly cools the sample to ~30 °C within 9–10 min, achieving an approximate quenching rate of 70 °C/min. Once cooled, we stop the Ar flow and extract the samples from the furnace. The reason behind this cooling routine is first to prevent any phase segregation or nucleation[3,5], and that if we attempt to cool the samples by extracting them from the furnace directly into air at 700 °C or higher, an oxidized surface layer forms, appearing as a spinel phase. Another option for achieving a similar quenching effect that could be implemented in future studies is to design a tube furnace that allows the boat to be moved from the hot zone to the edge of the tube while still under Ar flow. All synthesized high-entropy oxide ceramics by our methods have maintained their single-phase rock salt structure for months under ambient storage without detectable degradation, indicating good intrinsic stability.

For samples processed in air in Figure 3a and 5b, we follow similar preparation steps but sinter the samples in a Thermolyne box furnace under ambient conditions. We quench them by directly extracting the samples at 700 °C and cooling them in ambient $pO_2$. For the more stringent reduction experiments, such as the one in Fig. 5b, we sinter the samples in a tube furnace while flowing 100 SCCM of 1% $H_2$ forming gas, mixed from two Linde UHP cylinders (Ar and Ar + 5% $H_2$). We maintain the temperature at 1100 °C, achieving a $pO_2$ between $10^{-18}$–$10^{-20}$ atm.

### X-ray diffraction and X-ray fluorescence
We verify structural makeup for every single ceramic via X-ray diffraction (Panalytical Empyrean) using 2θ-θ Bragg-Brentano HD scans with a PIXcel3D detector and identify phases with PANalytical High-Score. We monitor equimolar cation compositions before and after sintering by X-ray fluorescence (Panalytical Epsilon 1) that we calibrate with in-house oxide standards.

### X-ray absorption fine structure
X-ray absorption fine structure (XAFS) spectra are collected using an easyXAFS300+ (Renton, WA) with a Ag X-ray tube operating at 35 kV and 25 mA[27]. Only the X-ray absorption near edge (XANES) region is measured with scans that ranged from 40 eV below to 175 eV above the respective absorption edge. The XAFS samples are massed to an appropriate amount determined via xraydb[28], a database used to calculate masses to use for transmission XAFS samples. These masses are

mixed with boron nitride as a filler so that a 5 mm diameter circular pellet could be made. A total of 12 scans are taken per edge per sample, the spectra are then merged and processed using the Demeter package for XAFS analysis[29].

### Electron microscopy and energy dispersive X-ray spectroscopy of MgCoNiZnMnFeO

The transmission electron microscopy (TEM) sample preparation is carried out with the Thermo Fisher Helios 660 NanoLab DualBeam focused ion beam (FIB). Selected area electron diffraction (SAED) experiments are performed using the Thermo Fisher Talos F200C at an accelerating voltage of 200 kV. Energy dispersive X-ray spectroscopy (EDS) elemental maps are acquired with the Thermo Fisher Talos 200×2, and a 1-pixel average post-filtering is applied.

### Thermodynamic analyses

Thermodynamic evaluations of the most stable oxidation states are carried out using the OpenCalphad software package[30], employing established thermodynamic data from assessments of relevant binary oxide systems. For each A–O system of interest (A = Mg, Co, Ni, Cu, Zn, Mn, Fe), we retrieve thermodynamic models and parameters for the stable phases from the literature[31–37]. We describe solution phases by the compound energy formalism[38], and we explicitly model stoichiometric compounds as functions of temperature.

We assess phase stabilities in pure $O_2$ environments under an ideal gas assumption, with total gas pressure ranging from $10^{-25}$ bar to $10^5$ bar. To ensure excess oxygen in every calculation, we set the elemental mole fractions $n(O) = 0.8$ and $n(A) = 0.2$. We determine phase-transition temperatures by minimizing the Gibbs free energy at various pressures, which enables us to construct phase boundaries over a range of −25 to 5 atm. We derive cation valences from the nominal stoichiometry of each phase, ignoring small off-stoichiometry effects in vacancy-tolerant solution phases. All calculations use the step-function capability in OpenCalphad[30], systematically exploring temperature–pressure space to identify equilibrium phase assemblies and oxidation states.

### Oxygen chemical potential overlap and bond length distribution descriptors

Oxygen chemical potential diagrams are constructed using the Materials Project GGA/GGA + U database as a more thorough number of calculations are present compared to the newer r²SCAN database[23] at the time of publication. Our oxygen chemical potential overlap, $\mu_{overlap}$, is defined for an $A_{1/2}B_{1/2}O$ stoichiometry as:

$$\mu_{overlap} = \min(A_2, B_2) - \max(A_1, B_1) \tag{1}$$

where $A_2$ and $B_2$ are the maximum oxygen chemical potential values for stable $A^{2+}$ and $B^{2+}$ regions, respectively, and $A_1$ and $B_1$ are the minimum oxygen chemical potential values for stable $A^{2+}$ and $B^{2+}$ regions, respectively. Note we can extend the $\mu_{overlap}$ definition for a five-component system (ABCDE)O as

$$\mu_{overlap} = \min(A_2, B_2, C_2, D_2, E_2) - \max(A_1, B_1, C_1, D_1, E_1) \tag{2}$$

A positive value for $\mu_{overlap}$ indicates an overlap in oxygen chemical potential space (a wider synthesis window), while a negative value indicates separation (a narrow/impossible synthesis window).

Values for $\Delta H_{mix}$ and $\sigma_{bonds}$ descriptors are adapted from our work in ref. 8 calculated using the CHGNet machine learning interatomic potential[17]. $\sigma_{bonds}$ represents the bond length distribution quantified using the standard deviation of relaxed first near-neighbor cation-anion bond lengths:

$$\sigma_{bonds} = \sqrt{\frac{\sum_i |a_i - \bar{a}|^2}{N}} \tag{3}$$

where $a_i$ is each bond length, $\bar{a}$ is the average bond lengths, and $N$ is the total number of bond lengths[8]

## Data availability

The data supporting the findings of this study are available within the paper and its Supplementary Information files. Source data are provided with this paper. Any additional datasets are available upon request from the corresponding author. Source data are provided with this paper.

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

## Acknowledgements

The authors gratefully acknowledge the full support from NSF MRSEC DMR-2011839. The authors also acknowledge Kaylin Lamaute and Billy Yang for their assistance with bulk ceramics processing, and Wes Auker along with the Penn State Materials Characterization Lab for their support in preparing the TEM samples. The authors further acknowledge many insightful discussions with the members of the Penn State NSF MRSEC IRG2: Crystalline Oxides with High Entropy.

## Author contributions

S.S.I.A. conceived the thermodynamics-inspired synthesis strategy, and the oxygen partial pressure control and chemical potential overlap hypotheses. S.S.I.A., M.F., and J.P.M. developed the experimental plan, built the experimental setup, and synthesized and characterized the ceramic pellets using XRD and XRF. J.P. and D.S. supported the ceramic synthesis. Y.T. and L.Q.C. performed thermodynamic analyses to construct the temperature and oxygen partial pressure phase diagrams. J.T.S. and S.B.S. developed the chemical potential overlap descriptor and performed the associated calculations. G.B. and C.M.R. performed the X-ray absorption investigations. S.V.G.A. and N.A. conducted the TEM investigations. S.S.I.A. analyzed and validated all data and wrote the original draft and subsequent revisions. S.S.I.A., M.F., D.S., J.T.S., and Y.T. designed the manuscript figures. J.P.M. provided project resources and supervision. All authors contributed to manuscript final review and editing.

## Competing interests

The authors declare no competing interests.
