## [Transparent Peer Review file · Nature Communications]

Thermodynamics-Inspired High-Entropy Oxide Synthesis

Corresponding Author: Dr Saeed Almishal

Version 0:

Reviewer comments:

Reviewer #1

(Remarks to the Author)

Dear Editor,

The manuscript reports a strategy to design and synthesize rock-salt structured high-entropy oxides (HEOs), highlighting the introduction of a new descriptor, oxygen chemical potential overlap, for predicting structural stability. This is of interest to the readers. The work is well-motivated, while the proposed approach for understanding the HEO phase stability is beyond traditional entropy-centric models. However, while the manuscript shows promise, I still have several concerns. I recommend that the manuscript be re-reviewed after a major revision. Detailed concerns are as follows,

- i. The structural information was provided via XRD patterns. The reviewer wonders if there is any trace secondary phase, and finds that some recent work reported the possible existence of nanoscale second phase oxides in HEOs. The authors should provide a more thorough microstructural characterization such as SEM-EDS and TEM-EDS.
- ii. In Figure 3, is Cu reduction thermodynamically inevitable? More discussion needs to be provided.
- iii. The authors claimed that “all measured cations in MgNiZnMnFeO and CoNiZnMnFeO—including Mn and Fe—‘predominantly’ exhibit a 2+ valence state” in XAS analysis, however the evidence is qualitative. The gap between non-quantitative descriptions and quantitative characterization results should be eliminated.
- iv. The interplay between proposed descriptor and existing descriptors is unclear. What combination of descriptors is more beneficial for predicting HEO structures?
- v. Despite the analysis using the proposed descriptor, why do some systems require reducing conditions? How to design atmosphere conditions from descriptors? How will the proposed descriptor work for the studies about synthesis using other techniques, e.g., magnetron sputtering, and joule heating?
- vi. Several existing descriptors (or descriptor combination) have shown potential to predict phase formation across HEO phases with different structures. Is the descriptor proposed in this work universal? Using it, is there potential to predict HEO structure in perovskites and other structures? There are some brief descriptions, but they are not clear. This part can be preliminarily supported by DFT data.
- vii. The reviewer encourages authors to provide a clear definition of σ bonds in the manuscript rather than relying on a preprint manuscript.

Reviewer #2

(Remarks to the Author)

Almishal et al. present a thermodynamically driven strategy for the synthesis of high-entropy oxides (HEOs) containing multivalent cations such as Fe^{2+} and Mn^{2+} , which are typically unstable under ambient conditions. The authors introduce a new descriptor, oxygen chemical potential overlap (μ overlap), to quantify the range of oxygen partial pressures over which all constituent cations remain in their desired oxidation states. Using this framework alongside conventional metrics such as mixing enthalpy and bond distortion, they predict and successfully synthesize seven single-phase rock salt HEOs under reducing conditions, bypassing the need for complex precursor chemistry.

This work goes beyond previous studies by generalizing the synthesis of Fe/Mn-containing HEOs. It shifts the field from empirical trial and error to rational design by combining machine learning, thermodynamic databases, and experimental validation. The μ overlap descriptor is the significant conceptual advance with the intent to enable targeted exploration of multicomponent oxide chemistries containing redox-sensitive elements.

Major Comments

1. 1100 °C is outside regions 2 and 3 in Figure 1b. So how do you use these conditions to experimentally probe regions 2 and 3 in the phase diagram? Are you using the 700 °C quench as the high temperature phase you are isolating? If so, region 3 is extremely narrow at that temperature (as we understand the paper, region 2 was used for synthesis). Pu et al.15 did not require a quench, why is it necessary in your method?

2. Were the starting reagents characterized? Particularly MnO and FeO are known to be unstable under ambient conditions. Pu et al.15 used a complex synthetic route to prevent their oxidation, while still employing pO₂ control during firing. Why was such an approach not necessary in your method?

3. We feel that the paper plays fast and loose with absolute generalizations of group's high/low ΔH and σ . For example, line 107 describes all blue compounds as having the lowest ΔH and σ , while there are green compounds with lower ΔH or σ .

Minor Comments

1. The SI Notes are not in a logical order

2. The paper seems to use the terms ambient pressure, ambient pO₂, ambient atmospheric pressure, etc. interchangeably. It would improve clarity if the paper used the same term throughout to refer to ambient pO₂.

3. Page 3 line 104-105 Co is repeated in the list

Reviewer #3

(Remarks to the Author)

Reviewer #4

(Remarks to the Author)

In this manuscript, Almishal et al, has explored the synthesis of high-entropy oxides (HEOs), specifically focusing on how controlling oxygen chemical potential can influence the stability and valence states of multivalent cations in rock salt structures. The research introduces a framework for predicting HEO stability and synthesizability based on thermodynamic stability and equilibrium analysis. The work is sound with in-depth mapping of component stability using computational tools corroborated by successful synthesis and in-depth characterization of multiple HEO compositions. There are few major concerns that needs to be addressed before it is in publishable form.

1. The authors introduce the "oxygen chemical potential overlap" as a descriptor for predicting stability and synthesizability and develop a phase diagram of temperature–oxygen partial pressure to identify regions (temperature and oxygen partial pressure overlap) where the cations are stable in the +2 oxidation state leading to stability in the rock salt structure.

However, a similar approach has already been demonstrated by Pu et al. in the literature (Pu et al., Sci. Adv. 9, eadi8809 (2023)), wherein they theoretically predict that different rocksalt HEOs can potentially be synthesized under low oxygen partial pressure using a similar temperature-partial pressure diagram. In this case, the authors should emphasize how there is a difference from this prior work, and whether this model offers improved and broader applicability.

2. While the work primarily focuses on enthalpic contributions, high-temperature solid-state synthesis often involves complex kinetic processes such as diffusion limitations, formation of intermediate phases, and metastable products. Particularly for the experimental part, the authors should acknowledge the role of kinetics and the possibility/difficulty of including these as descriptors in the model.

3. The model assumes ideal random mixing of cations. Experimental and computational studies increasingly show that short-range ordering plays a significant role in determining the stability and properties of HEOs. Can the authors comment on how a deviation from ideal random mixing of the rock salt binary oxides can affect their thermodynamic calculations? Alternatively, were they able to observe any short-range ordering of cations in their experimental studies?

4. Have the author considered examining the long term stability of the synthesised HEO compositions specifically in harsh chemical environments or elevated temperatures or pressures.

5. "Although we focus on rock salt HEOs, our methods are chemically and structurally agnostic, providing a broadly adaptable framework for navigating HEO thermodynamics and enabling a broader compositional range with contemporary property interest." This statement in the abstract is vague, especially the use of the word agnostic. While the theoretical model and its experimental validation for rock salt HEOs are noteworthy, it is unclear how the framework can be readily extended to other complex oxide systems such as perovskites, spinels, or layered oxides. These structures are more complex and can involve different structural constraints and coordination environments. Can the authors comment on the generalizability of the model, with particular emphasis on what factors and structural descriptors need to be introduced when working on more complex phases?

6. The legends in most of the figures (Fig. 2, 3c, 5) have quite small font sizes. It would be helpful if authors can increase the font sizes and make it clear.

Reviewer #5

(Remarks to the Author)

Version 1:

Reviewer comments:

Reviewer #1

(Remarks to the Author)

My concerns have been addressed. I am pleased to recommend its publication.

Reviewer #2

(Remarks to the Author)

I recommend this manuscript for acceptance.

Reviewer #4

(Remarks to the Author)

After going through all the comments from different reviewers and the response from the authors, I would recommend publication of this manuscript. The revised manuscript has improved significantly. I have no further comments on the manuscript.

Reviewer #5

(Remarks to the Author)

We sincerely thank the reviewers and editors for their thoughtful feedback and careful evaluation of our work. We believe their insights have helped us strengthen the manuscript, sharpen our arguments, and reinforce the support for our conclusions and the broader impact. Below, we provide a detailed, point-by-point response to all comments. Revisions are highlighted in color in the main manuscript and annotated to directly link each change to the corresponding reviewer feedback for ease of review.

Reviewers' comments

Reviewer #1 (Remarks to the Author):

The manuscript reports a strategy to design and synthesize rock-salt structured high-entropy oxides (HEOs), highlighting the introduction of a new descriptor, oxygen chemical potential overlap, for predicting structural stability. This is of interest to the readers. The work is well-motivated, while the proposed approach for understanding the HEO phase stability is beyond traditional entropy-centric models. However, while the manuscript shows promise, I still have several concerns. I recommend that the manuscript be re-reviewed after a major revision.

We sincerely thank the reviewer for their thoughtful and constructive feedback. We appreciate the recognition of the significance of our work in proposing a new descriptor, oxygen chemical potential overlap (μ_{overlap}), to advance the understanding of high-entropy oxide (HEO) phase stability beyond traditional entropy-centric models.

In response to the reviewer's concerns, we have undertaken substantial revisions to the manuscript. These include expanded discussions, additional experimental validations, and a clearer articulation of the interplay between μ_{overlap} and other key descriptors. We have also addressed specific requests for more rigorous microstructural and spectroscopic characterizations, provided quantitative XAS analyses, and clarified the descriptor's applicability across different structures and synthesis techniques. Detailed responses to each of the reviewer's comments are provided below.

Detailed concerns are as follows,

i. The structural information was provided via XRD patterns. The reviewer wonders if there is any trace secondary phase and finds that some recent work reported the possible existence of nanoscale second phase oxides in HEOs. The authors should provide a more thorough microstructural characterization such as SEM-EDS and TEM-EDS.

We thank the reviewer for this important suggestion. We believe this significantly strengthens the experimental validation of the work. In response, we have conducted comprehensive microstructural characterization to confirm the single-phase nature of the parent six component composition MgCoNiZnMnFeO . Specifically, we have added TEM-selected area electron diffraction (SAED) patterns and TEM-EDS elemental mapping at three magnifications (50 nm, 100 nm, and 500 nm) to the manuscript in Figure 5 (also shown below) and in Supporting Information Note 5. Additionally, we now include a complete set of XANES spectra for the parent

Figure 5: (a) X-ray diffraction patterns of the 6-component high-entropy composition $Mg_{1/6}Co_{1/6}Ni_{1/6}Zn_{1/6}Mn_{1/6}Fe_{1/6}O$ ($MgCoNiZnMnFeO$) sintered for 5 hours under 100SCCM of Ar at different temperatures, suggesting that the transition to single phase occurs between 850-900°C with the disappearance of the wurtzite (W) peaks. (b) X-ray diffraction patterns of $MgCoNiZnMnFeO$ sintered at 1100°C under ambient pO_2 , optimized reducing conditions and excess reducing conditions; while it exhibits single phase rock salt when sintered for 5 hours under 100 SCCM flow of Ar, a metallic phase emerge when either small percentage of H_2 is added or if sintered for prolonged time periods. ‘RS’ denotes peaks from the rock salt structure, ‘S’ denotes those from the cubic spinel structure, ‘W’ indicates the wurtzite structure, and ‘FCC’ indicates the face-centered cubic metallic structure. (c) Selected area electron diffraction (SAED) along the [110] zone axis of our parent $MgCoNiZnMnFeO$ sintered for 5 hours under 100SCCM Ar, with the inset showing the selected area (yellow circle ~700 nm in radius) corresponding to the diffraction pattern. The electron diffraction pattern is consistent with the rock salt crystal structure. (d) Energy-dispersive spectroscopy (EDS) maps showing a homogeneous distribution of elements at the 50 nm scale. (e) Mn K-edge and Fe K-edge photon energy vs valence state with best-fit line confirming a predominance of Mn^{2+} and Fe^{2+} within $MgCoNiZnMnFeO$ composition, with the 2+ reference values are indicated by orange stars.

six-component composition in Note 3 and summarize the main results in Figure 5. Together, these results rigorously confirm the formation of a single-phase rock salt structure, with no detectable secondary phases in our measurements.

ii. In Figure 3, is Cu reduction thermodynamically inevitable? More discussion needs to be provided.

We thank the reviewer for their question. In the revised manuscript, we have expanded the discussion in response to this comment. Specifically, we note that deviations from Region 1, either toward higher temperatures or lower pO_2 , lead to inevitable CuO reduction and Cu melting under equilibrium synthesis conditions, as reflected by the Cu liquidus line being the only liquidus

boundary shown in Figure 1(b). This thermodynamic framework indicates that, under such conditions, Cu reduction is favored. However, we emphasize that the thermodynamic diagrams present only the equilibrium thermodynamic factors. In practice, whether Cu reduction occurs also depends on kinetic factors, such as the activation energy of the reduction reaction and oxygen diffusion within the solid. These kinetic barriers can significantly slow down reduction, meaning that the actual Cu average valence is highly sensitive to the details of the synthesis process, especially the sintering temperature and dwell time. Therefore, while Cu reduction is thermodynamically inevitable under sufficiently reducing conditions, kinetic limitations and multicomponent interactions can modulate the reduction pathway and must be carefully managed through synthesis design.

iii. The authors claimed that “all measured cations in MgNiZnMnFeO and CoNiZnMnFeO—including Mn and Fe—‘predominantly’ exhibit a 2+ valence state” in XAS analysis, however the evidence is qualitative. The gap between non-quantitative descriptions and quantitative characterization results should be eliminated.

We thank the reviewer for their comment. In our original analysis, we extracted the K-edge energies from the second derivative of the XANES spectra and compared them to standard reference values for known oxidation states. We refrained from assigning a single definitive oxidation state value in the main text for the high entropy oxides because such estimates rely on best linear fits, which inherently involve some degree of model dependence. However, as the reviewer points out, the absence of explicit quantitative values could be seen as limiting. Therefore, to strengthen our arguments and conclusions, we have now provided a full table in the Supporting Information Note 3 listing the exact K-edge energies, the corresponding oxidation state from the standard references, and the best linear fit-derived oxidation states for Mn and Fe in each high-entropy composition. These results consistently show that the cations are predominantly in the 2+ state, reinforcing the conclusions without ambiguity. Additionally, the updated manuscript now includes XAS analysis for the six-component parent composition.

iv. The interplay between proposed descriptor and existing descriptors is unclear. What combination of descriptors is more beneficial for predicting HEO structures?

We thank the reviewer for highlighting this important point and apologize for not making it more explicit in the original manuscript. Based on our studies over the past decade and insights from the broader community, we find that no single descriptor is sufficient to fully predict the stability and synthesizability of HEOs. A combination of descriptors is essential. Specifically, we believe that enthalpy of mixing (ΔH_{mix}), bond length distribution (σ_{bonds}), oxygen chemical potential overlap (μ_{overlap}), and binary phase diagram information form a comprehensive set necessary to reliably predict both phase stability and synthesizability. We have now clarified this in the revised manuscript, emphasizing how these descriptors complement each other in capturing the complex thermodynamic factors that govern HEO formation. One example of those edits is included below;

“Therefore, for accurate HEO stability predictions, it is critical to consider μ_{overlap} alongside conventional descriptors^{8,16}. Specifically, a comprehensive predictive framework for realizing equilibrium HEO phases should, at a minimum, incorporate ΔH_{mix} , σ_{bonds} and μ_{overlap} , together with insights from binary and higher-order phase diagrams and decomposition enthalpy^{3,8}. Future studies should further incorporate finite-temperature and configurational entropy effects, moving beyond the ideal mixing assumption, further enhancing stability and synthesizability predictions.”

v. Despite the analysis using the proposed descriptor, why do some systems require reducing conditions? How to design atmosphere conditions from descriptors? How will the proposed descriptor work for the studies about synthesis using other techniques, e.g., magnetron sputtering, and joule heating?

We thank the reviewer for these insightful questions. The proposed μ_{overlap} descriptor is fundamentally an equilibrium thermodynamic metric, highlighting the conditions under which the desired cation oxidation states can be stabilized relative to the ambient oxygen chemical potential. In cases where certain cations, such as Mn or Fe, favor higher oxidation states under ambient conditions, reducing the oxygen partial pressure shifts the chemical potential, stabilizing lower oxidation states and thereby favoring the formation of a single-phase rock salt structure. This explains why some systems require reducing atmospheres during synthesis.

Regarding the design of synthesis conditions, μ_{overlap} provides a framework for estimating the oxygen chemical potential range needed to stabilize the targeted phase. By comparing chemical potential overlaps under varying environmental conditions, one can qualitatively predict whether reducing, inert, or oxidizing atmospheres are required. A full quantitative prediction of the optimal atmosphere would require extending the descriptor to account for temperature effects, which we now explicitly include in Supporting Information Note 7, with a detailed solution for the case of incorporating Mn and Fe following the ideal gas approximation.

For far-from-equilibrium synthesis techniques such as magnetron sputtering, joule heating, or pulsed laser deposition (PLD), the descriptor still offers valuable guidance. Although these methods operate under kinetic constraints, local thermodynamic valence and oxidation states parameters captured by μ_{overlap} remain relevant. It is important to note, however, that far-from-equilibrium syntheses often occur at temperatures or under conditions where equilibrium single-phase formation is not thermodynamically favored. For example, in our previous study on MgCoNiCuZnO thin films grown by PLD, Co^{3+} was observed to form, which can be rationalized by considering that Co^{3+} overlaps with the 2^+ oxidation states at lower synthesis temperatures based on μ_{overlap} maps. To fully capture these effects, binary phase diagrams must also be considered alongside μ_{overlap} , especially when growth temperatures deviate significantly from equilibrium conditions. Additionally, far-from-equilibrium synthesis techniques enable access to different valence environments, local structures, and order evolution pathways that are not accessible under

equilibrium conditions, further complicating the prediction landscape. To address these complexities, we have been utilizing phase-field modeling approaches in our recent work, allowing us to simulate the evolution of local structure and valence states during non-equilibrium growth. To clarify these connections, we have added a detailed Note 9 in the Supporting Information discussing the relationship between equilibrium-based synthesis predictions and far-from-equilibrium synthesis strategies, highlighting the complementary role of μ_{overlap} , phase diagrams, and phase-field models.

vi. Several existing descriptors (or descriptor combination) have shown potential to predict phase formation across HEO phases with different structures. Is the descriptor proposed in this work universal? Using it, is there potential to predict HEO structure in perovskites and other structures? There are some brief descriptions, but they are not clear. This part can be preliminarily supported by DFT data.

We thank the reviewer for this important comment, which helps strengthen our work. We agree that universality is a critical requirement for any proposed descriptor. In the revised manuscript, we have expanded the discussion and provided additional examples with supporting synthesis results to demonstrate that the oxygen chemical potential overlap (μ_{overlap}) descriptor is applicable beyond the rock salt structure. Specifically, we extend the application of μ_{overlap} to rare-earth sesquioxides and defective fluorites and discuss its relevance to more complex structures such as perovskites by independently treating each cation sublattice according to its preferred valence. These additions are now clearly and explicitly included in the main text and Supporting Information.

As the reviewer also noted in Comment 4, similar to other descriptors proposed in the literature, we acknowledge that μ_{overlap} alone cannot fully predict phase formation. It must be used in combination with other key descriptors, such as enthalpy of mixing (ΔH_{mix}), bond length distribution (σ_{bonds}), and insights from binary and higher-order phase diagrams.

Regarding the suggestion for DFT validation, we would like to highlight that in our related work (*Discovering High-Entropy Oxides with a Machine-Learning Interatomic Potential*, *Phys. Rev. Lett.* **134**, 216101 (2025)), we demonstrated a one-to-one comparison between DFT calculations and machine-learning interatomic potentials. While machine-learning models do not replace DFT, they leverage extensive DFT databases, enabling rapid exploration of vast compositional spaces with near-DFT accuracy at a fraction of the cost. This provides strong support for the thermodynamic principles and open-source DFT database underlying the μ_{overlap} framework.

vii. The reviewer encourages authors to provide a clear definition of σ_{bonds} in the manuscript rather than relying on a preprint manuscript.

We thank the reviewer for this helpful suggestion. We have included a clear and explicit definition of σ_{bonds} directly in the manuscript in the computational methods to ensure clarity and self-containment, as suggested. We have additionally now updated the reference to our work, which has been published in *Physical Review Letters* (DOI: <https://doi.org/10.1103/PhysRevLett.134.216101>).

Reviewer #2 (Remarks to the Author):

Almishal et al. present a thermodynamically driven strategy for the synthesis of high-entropy oxides (HEOs) containing multivalent cations such as Fe^{2+} and Mn^{2+} , which are typically unstable under ambient conditions. The authors introduce a new descriptor, oxygen chemical potential overlap (μ_{overlap}), to quantify the range of oxygen partial pressures over which all constituent cations remain in their desired oxidation states. Using this framework alongside conventional metrics such as mixing enthalpy and bond distortion, they predict and successfully synthesize seven single-phase rock salt HEOs under reducing conditions, bypassing the need for complex precursor chemistry.

This work goes beyond previous studies by generalizing the synthesis of Fe/Mn-containing HEOs. It shifts the field from empirical trial and error to rational design by combining machine learning, thermodynamic databases, and experimental validation. The μ_{overlap} descriptor is the significant conceptual advance with the intent to enable targeted exploration of multicomponent oxide chemistries containing redox-sensitive elements.

We sincerely thank the reviewer for their thoughtful and constructive feedback, as well as for recognizing the significance of our work in advancing the rational synthesis of redox-sensitive high-entropy oxides. We appreciate the recognition of the μ_{overlap} descriptor as a conceptual advance and of the broader framework combining thermodynamic analysis, machine learning, and experimental validation. In response to the reviewer's comments, we have carefully revised the manuscript to address all major and minor concerns. Detailed responses to each of the reviewer's comments are provided below.

Major Comments

1. 1100 °C is outside regions 2 and 3 in Figure 1b. So how do you use these conditions to experimentally probe regions 2 and 3 in the phase diagram? Are you using the 700 °C quench as the high temperature phase you are isolating? If so, region 3 is extremely narrow at that temperature (as we understand the paper, region 2 was used for synthesis). Pu et al.¹⁵ did not require a quench, why is it necessary in your method?

We thank the reviewer for raising this important point and apologize for any confusion caused by the original figure. Regions 2 and 3 were initially capped at the Cu liquidus line, but in fact, both

extend up to 1200 °C. We have updated the phase diagram and clarified the captions accordingly in the revised manuscript.

Regarding the synthesis conditions, the samples were fired at 1100 °C, which falls within the extended Region 2 once the temperature dependence of the oxygen chemical potential is considered. Higher temperatures expand the window where the desired +2 oxidation states are thermodynamically stable. We also note that the phase diagram presented is based on binary systems; in practice, synthesis could occur at any point between the onset of Region 2 and the end of Region 3, as we now clarify in the text.

The quenching step is essential to preserve the oxidation states and maintain the single-phase rock salt structure by minimizing reoxidation, suppressing short-range ordering, and preventing the formation of lower-enthalpy competing phases during cooling. Without quenching, slow cooling would promote Fe²⁺ and Mn²⁺ reoxidation and secondary phase formation. Quenching also helps retain the high-entropy random cation distribution established at high temperatures. This is particularly critical for our compositions, which include Zn²⁺, a known destabilizer of the rock salt phase, making them more challenging to stabilize than the systems studied by Pu et al.

In comparison, Pu et al. employed carefully engineered precursors and controlled oxygen environments during synthesis and cooling, introducing 90 mg of MnO₂ as an oxygen generator. However, they did not report the overall oxygen partial pressure in the furnace, making direct comparison to our method difficult. For example, in their Supporting Information, they describe annealing a sample for 1 hour at 1000 °C with external O₂, resulting in the coexistence of spinel and metallic phases. They attribute this to limited oxygen diffusion within the fixed bed, leading to an over-oxidized top layer and an under-oxidized bottom layer, an issue they suggest can be mitigated by prolonging the annealing time. By contrast, our approach uses a simpler and more broadly accessible method: firing under a 50 SCCM Ar flow without the need for external O₂ sources, followed by quenching under the same atmosphere. These synthesis steps are now detailed explicitly in the revised Methods section and further clarified in the revised manuscript body.

2. Were the starting reagents characterized? Particularly MnO and FeO are known to be unstable under ambient conditions. Pu et al.¹⁵ used a complex synthetic route to prevent their oxidation, while still employing pO₂ control during firing. Why was such an approach not necessary in your method?

We thank the reviewer for this important question. To ensure the integrity of our starting materials, we characterized the commercially sourced MnO and FeO powders by X-ray diffraction (XRD) prior to use. The patterns confirmed the expected rock salt structure with no secondary phases, as now noted in the Methods and shown in Supporting Information Note 1 and included below for reference. To minimize oxidation, we vacuum-sealed the powders; notably, even samples stored

Figure S1. (a) X-ray diffraction (XRD) patterns of MnO and FeO as-received starting precursors

under ambient conditions retained their structure over extended periods, demonstrating adequate stability.

Unlike Pu et al.¹⁵, who employed engineered precursors and oxygen control, our method uses simple commercially available oxide precursors without complex synthetic routes. High-temperature firing under controlled atmospheres re-equilibrates and maintain cation oxidation states, stabilizing Fe^{2+} and Mn^{2+} during synthesis. We believe this simplicity is a key advantage, enabling accessible and scalable synthesis of redox-sensitive high-entropy oxides. These clarifications have been incorporated into the revised Methods section.

3. We feel that the paper plays fast and loose with absolute generalizations of group's high/low ΔH and σ . For example, line 107 describes all blue compounds as having the lowest ΔH and σ , while there are green compounds with lower ΔH or σ .

We thank the reviewer for their comment and bringing this to our attention. We have clarified that these compositions have the lowest ΔH_{mix} and σ_{bonds} descriptor values for the 5- and 6-cation rock salt HEO combinations. We note that there are indeed 4-cation combinations that possess descriptor values less than these seven compositions explored here as can be observed in Figure 1a. To address this, we have revised the language in the main text for greater precision, added a reference to the full descriptor table provided in the Supporting Information Table S3, and updated the figure caption to clarify this point.

Minor Comments

1. The SI Notes are not in a logical order

We thank the reviewer for this helpful suggestion. In response, we have reorganized and extended the Supporting Information Notes to improve their logical flow and readability. If the reviewer has specific suggestions for further reordering, we would be happy to consider them.

2. The paper seems to use the terms ambient pressure, ambient pO₂, ambient atmospheric pressure, etc. interchangeably. It would improve clarity if the paper used the same term throughout to refer to ambient pO₂.

We thank the reviewer for pointing this out. To improve clarity and consistency, we have revised the manuscript to uniformly use the term *ambient pO₂* when referring to the oxygen partial pressure under standard atmospheric conditions.

3. Page 3 line 104-105 Co is repeated in the list

We thank the reviewer for catching this error. We have corrected the repeated listing of Co.

Reviewer #3 (Remarks to the Author):

We sincerely thank the reviewer and their co-reviewer for their time and thoughtful evaluation of our work. We appreciate their efforts and the important role of co-review initiatives in supporting the next generation of researchers and ensuring a thorough and constructive peer-review process.

Reviewer #4 (Remarks to the Author):

In this manuscript, Almishal et al, has explored the synthesis of high-entropy oxides (HEOs), specifically focusing on how controlling oxygen chemical potential can influences the stability and valence states of multivalent cations in rock salt structures. The research introduces a framework for predicting HEO stability and synthesizability based on thermodynamic stability and equilibrium analysis. The work is sound with in-depth mapping of component stability using computational tools corroborated by successful synthesis and in-depth characterization of multiple HEO compositions. There are few major concerns that needs to be addressed before it is in publishable form.

We sincerely thank the reviewer for their thoughtful and constructive feedback, as well as for recognizing the significance and depth of our work. We have carefully addressed all major concerns raised, expanded the manuscript where necessary, and provided additional experimental validation and detailed discussions to strengthen the framework we propose. The specific responses to each comment are detailed below.

1. The authors introduce the “oxygen chemical potential overlap” as a descriptor for predicting stability and synthesizability and develop a phase diagram of temperature–oxygen partial pressure to identify regions (temperature and oxygen partial pressure overlap) where the cations are stable in the +2-oxidation state leading to stability in the rock salt structure. However, a similar approach has already been demonstrated by Pu et al. in the literature (Pu et al., *Sci. Adv.* 9, eadi8809 (2023)), wherein they theoretically predict that different rock salt HEOs can potentially be synthesized under low oxygen partial pressure using a similar temperature-partial pressure diagram. In this case, the authors should emphasize how there is a difference from this prior work, and whether this model offers improved and broader applicability.

We thank the reviewer for their comment. We agree that we should more carefully highlight the improvements of our model over those documented by Pu et al. Compared to focusing solely on the Mn and Fe cases as done in Pu et al., our methodology here (i.e., the CALPHAD phase diagram as well as the oxygen chemical potential overlap descriptor) explores how changes in oxygen chemical potential affect all metal cation components within a HEO composition. Our proposed framework therefore captures the difficulty (and perhaps impossibility) for Mn/Fe with Cu to coexist as 2+ cations. This dissimilarity in cation valence stability is particularly evident in Figure 3b, where MgCoNiCuZnO synthesis attempts under argon result in a multiphase material containing FCC metal corresponding to elemental Cu. HEO compositions containing Mn or Fe *without* Cu, however, all result in single-phase rock salt oxides under the same conditions. This dissimilarity for Mn/Fe + Cu compositions is not captured within Pu et al.'s work. Importantly, our proposed oxygen chemical potential overlap descriptor enables us to quantify the cation valence compatibility observed in the CALPHAD phase diagram for entire HEO compositions in a computationally lightweight manner.

While our focus in this manuscript is on the chemical potential overlap descriptor (μ_{overlap}), we note that Pu et al.'s computational descriptors do not include any quantification of local distortion or size disparity, which are well-known to have significant influence on solid-solution formation (e.g., Hume-Rothery or Pauling's rules). Our σ_{bonds} descriptor quantifies these size differences by utilizing relaxed supercells. We additionally note that while our ΔH_{mix} descriptor is inherently similar to Pu et al.'s mixing enthalpy, we compute these values for the full HEO composition rather than approximating from two-component compositions. We believe that this approach is more representative of the complex HEO thermodynamic landscape, especially when charge compensation between different cations is possible.

Additionally, while Pu et al. focused exclusively on a single Fe- and Mn-containing system without Zn, we demonstrate the predictive power of μ_{overlap} across multiple compositions and provide experimental validation through the successful synthesis of seven different high-entropy oxides. We have further extended the applicability of μ_{overlap} to more complex systems such as rare-earth sesquioxides: bixbyite and defective fluorites, and we outline how it can be adapted to other structures such as perovskites by considering sublattice-specific valence preferences. These clarifications have now been incorporated into the revised manuscript and Supporting Information to emphasize the broader applicability and conceptual advances provided by the combined ΔH_{mix} , σ_{bonds} , and μ_{overlap} framework.

2. While the work primarily focuses on enthalpic contributions, high-temperature solid-state synthesis often involves complex kinetic processes such as diffusion limitations, formation of intermediate phases, and metastable products. Particularly for the experimental part, the authors should acknowledge the role of kinetics and the possibility/difficulty of including these as descriptors in the model.

We thank the reviewer for this insightful comment. We fully agree that high-temperature solid-state synthesis involves complex kinetic processes, including diffusion limitations and intermediate phase formation. Our work primarily focuses on developing a thermodynamic framework to identify favorable synthesis windows under equilibrium conditions. However, thermodynamic favorability alone does not guarantee phase formation; kinetic factors are critical, particularly for high-entropy oxides where metastable phases are often stabilized through kinetic trapping. To address these complexities, our synthesis strategy includes a five-hour hold at 1100 °C to allow sufficient equilibration, followed by rapid quenching, as now detailed in the revised Methods section. This approach minimizes reoxidation, suppresses short-range ordering, and prevents secondary phase formation. The effectiveness of this strategy is supported by our newly added TEM-EDS analysis in Figure 5 and Note 5 as also discussed in response to Reviewer 1, which shows a homogeneous cation distribution with no observed clustering.

While μ_{overlap} is a thermodynamic descriptor and does not explicitly incorporate kinetic factors, we have added a new Supporting Information Note discussing kinetic influences. Regarding the possibility of including descriptors that capture kinetic effects, we now include in the Supporting Information how phase-field modeling and decomposition enthalpy can assist in identifying metastable products and local nanostructure formation. We have also expanded the manuscript to place μ_{overlap} in the broader context of ΔH_{mix} , σ_{bonds} , and kinetic effects. Additionally, we now demonstrate the broader applicability of μ_{overlap} by extending it to rare-earth sesquioxides (bixbyite structure) and defective fluorites, supported by new synthesis results. These examples highlight that despite kinetic complexities, μ_{overlap} reliably contributes to guiding the synthesis of structurally and chemically diverse oxides.

3. The model assumes ideal random mixing of cations. Experimental and computational studies increasingly show that short-range ordering plays a significant role in determining the stability and properties of HEOs. Can the authors comment on how a deviation from ideal random mixing of the rock salt binary oxides can affect their thermodynamic calculations? Alternatively, were they able to observe any short-range ordering of cations in their experimental studies?

We thank the reviewer for this important comment. In our thermodynamic model, although not stated explicitly, we assume ideal random mixing of cations, a common first approximation in HEOs. However, we recognize that short-range ordering can influence phase stability and functional properties. We have now added Supporting Information Note 9 to provide a detailed discussion of these effects and acknowledged this limitation in the main manuscript. In the newly added Supporting Information Note, we discuss how phase-field modeling and decomposition enthalpy can predict order evolution and highlight recent work using cluster expansion techniques and first-principles calculations. These approaches represent promising pathways to extend the overall predictive framework for HEO synthesis. Additionally, to probe for ordering, we have added high-resolution TEM-EDS mapping at multiple length scales. These measurements show

no evidence of cation clustering within the resolution limits, supporting the assumption of a predominantly random cation distribution.

4. Have the author considered examining the long-term stability of the synthesized HEO compositions specifically in harsh chemical environments or elevated temperatures or pressures.

We thank the reviewer for their question. All synthesized high-entropy oxide ceramics have maintained their single-phase rock salt structure for over eight months under ambient storage without detectable degradation, indicating good intrinsic stability. This observation is now noted in the revised manuscript. While a systematic evaluation under harsh chemical environments, elevated temperatures, or pressures is beyond the scope of this study, we agree that this is an important direction for future work.

5. “Although we focus on rock salt HEOs, our methods are chemically and structurally agnostic, providing a broadly adaptable framework for navigating HEO thermodynamics and enabling a broader compositional range with contemporary property interest.” This statement in the abstract is vague, especially the use of the word agnostic. While the theoretical model and its experimental validation for rock salt HEOs are noteworthy, it is unclear how the framework can be readily extended to other complex oxide systems such as perovskites, spinels, or layered oxides. These structures are more complex and can involve different structural constraints and coordination environments. Can the authors comment on the generalizability of the model, with particular emphasis on what factors and structural descriptors need to be introduced when working on more complex phases?

We thank the reviewer for highlighting this important point. We have carefully reconsidered the wording and agree that additional context is necessary to avoid ambiguity. In the revised manuscript, we now provide a clearer explanation of the intended generalizability of the framework. To demonstrate this broader applicability, we have extended μ_{overlap} beyond rock salt structures to more complex systems. Specifically, in the revised manuscript and the new Supporting Information Note, we present two additional examples: rare-earth sesquioxides (bixbyite structure) and defective fluorites, where μ_{overlap} successfully guides synthesis, supported by new experimental validation. Figure S8 is added below for reference.

Furthermore, our three proposed descriptors, enthalpy of mixing (ΔH_{mix}), bond length distribution (σ_{bonds}), and oxygen chemical potential overlap (μ_{overlap}), are thermodynamically based quantities derived from machine-learning interatomic potentials or open-source DFT databases. They quantify the enthalpic barrier to single-phase formation, local lattice distortion from ionic size differences or distortions, and valence compatibility with respect to available oxygen content, respectively. These thermodynamic quantities remain applicable to more complex crystal structures such as perovskites, spinels, and layered oxides. For structures with multiple symmetrically unique cation sites, as in perovskites and spinels, the σ_{bonds} descriptor can be

Figure S9. (a) Oxygen chemical potential diagrams for Y-, La-, Pr-, and Sm-oxides, highlighting the stability windows of their $(A^{3+})_2O_3$ binaries, and for ZrO_2 , reflecting its 4+ stability. Ce–O is shown separately to capture its multiple accessible oxidation states. Green shaded regions indicate oxygen chemical potential ranges where the desired stoichiometries are thermodynamically stable. All data are extracted from the Materials Project database. (b) μ_{overlap} heat map for all two-cation combinations, illustrating the range of overlap values for Ce depending on its oxidation state, and the strong compatibility between Zr^{4+} and both Ce^{4+} and the trivalent cations. (c) X-ray diffraction (XRD) patterns of $(Ce,La,Pr,Sm,Y)_2O_3$ (bixbyite) and $Zr(Ce,La,Pr,Sm,Y)O_2$ (defective fluorite) compositions. Samples were sintered at 1400 °C after 18 hours of wet milling with 3 mm YSZ balls and methanol.

extended by evaluating bond length distortions for each distinct cation coordination environment. While this increases the dimensionality of the descriptor, smaller σ_{bonds} values across all sites should still correlate with a tendency toward single-phase formation. Additionally, compositional asymmetry between cation sites, especially in perovskites, can be captured by extending our framework like established tolerance factor-based approaches (e.g., Bartel et al., *Sci. Adv.*, 5, eaav0693 (2019), DOI:10.1126/sciadv.aav0693). Importantly, the ΔH_{mix} and μ_{overlap} descriptors are composition-specific and therefore largely insensitive to the targeted crystal structure, providing further confidence in the framework’s applicability to a wide range of high-entropy oxide chemistries. These clarifications, along with the new experimental extensions, have been incorporated into the revised manuscript and Supporting Information to provide a more transparent and robust explanation of the framework’s generalizability, while preserving the original terminology.

6. The legends in most of the figures (Fig. 2, 3c, 5) have quite small font sizes. It would be helpful if authors can increase the font sizes and make it clear.

We thank the reviewer for this helpful suggestion. We have revised the figures and prepared higher-resolution versions with clearer, larger fonts. Due to file size limitations and resolution degradation in Microsoft Word files, we will provide the final high-resolution figures separately when prompted during the revision stages to ensure optimal clarity.

Reviewer #5 (Remarks to the Author):

We sincerely thank the reviewer and their co-reviewer for their time and thoughtful evaluation of our work. We appreciate their efforts and the important role of co-review initiatives in supporting the next generation of researchers and ensuring a thorough and constructive peer-review process.